# Behaviour and Welfare Impacts of Releasing Elephants from Overnight Tethers: A Zimbabwean Case Study

**DOI:** 10.3390/ani12151933

**Published:** 2022-07-29

**Authors:** Ellen Williams, Natasha Clark, Jake Rendle-Worthington, Lisa Yon

**Affiliations:** 1Department of Animal Health, Behaviour and Welfare, Harper Adams University, Newport TF10 8NB, UK; ewilliams@harper-adams.ac.uk; 2School of Veterinary Medicine and Science, The University of Nottingham, Sutton Bonington LE12 5RD, UK; natasha.clark@liverpool.ac.uk; 3eleCREW, Zimbabwe reg Trust, Jafuta Reserve, Victoria Falls 999135, Zimbabwe; jake@weareallmammals.org; 4We Are All Mammals, CIO Reg No. 1184219, Broomfield TA5 2EQ, UK

**Keywords:** elephants, welfare, activity budgets, semi-captive, tethered

## Abstract

**Simple Summary:**

In the southern African elephant tourism industry, many elephants are routinely chained or tethered for prolonged periods of time, particularly overnight. There are many negative implications of such restrictive management on elephant welfare. In this pilot study, a group of four male semi-captive African elephants at a tourist facility in Zimbabwe were being taken off overnight tethers and put into small pens, as a strategy to improve elephant welfare. Behavioural data were collected when elephants were on tethers, and approximately four weeks and eight weeks after removal of tethers. Behavioural changes were noted, after removal of tethers, which were indicative of improved welfare (e.g., increased lying rest, increased positive social behaviour and reduced abnormal repetitive behaviours). Importantly, there were no significant increases in aggression (either to their human handlers or to other elephants) following this change in management. To improve elephant welfare in southern African tourism facilities, we strongly advocate management practices which enable greater choice and freedom of movement overnight, which includes ceasing the use of overnight tethers and provision of opportunities for physical interaction with other elephants overnight.

**Abstract:**

Within the southern African elephant tourism industry, chaining or tethering elephants is still a relatively routine practice, despite the known negative impacts. Cited reasons for chaining include fear of aggressive interactions between elephants when handlers are absent, or a general increase in expression of aggressive behaviours (both to other elephants and to their human handlers). In Zimbabwe, concerns expressed include the danger of elephants escaping and entering human-inhabited areas. Four male semi-captive elephants at a Zimbabwe tourist facility were taken off overnight (~12 h) tethers and were placed in small pens (‘bomas’), approximate sizes from 110 m^2^ to 310 m^2^), as part of a strategy to improve elephant welfare. Behavioural data were collected from overnight videos from December 2019 to March 2020, between 18:00 to 06:00, using focal, instantaneous sampling (5-min interval). Data were collected for three nights at three time periods: (i) Tethered; (ii) approximately four weeks post-release; (iii) approximately eight weeks post-release. Behavioural change over these time points was analysed using general linear models with quasibinomial error structures. Behavioural changes indicative of improved welfare were observed following these management changes, and no significant increases in aggression were observed either between elephants, or towards their human handlers. Proportion of time engaging in lying rest was higher in the first month after release from tethering (mean ± SD, 50 ± 14%) than when elephants were tethered (20 ± 18%) (*p* < 0.05). Additionally, although not statistically significant, stereotypies were reduced when elephants were no longer tethered (4 ± 6% observations tethered compared to 2 ± 2% off tethers), and positive social behaviour also increased (1 ± 1% on tethers, 2 ± 2% off tethers), with the greatest improvements seen in the pair-housed elephants. To improve elephant welfare in southern African tourism facilities we strongly advocate that less restrictive management practices which enable greater choice and freedom of movement overnight are implemented.

## 1. Introduction

Tourism is estimated to represent 10% of global gross domestic product (GDP) and has been highlighted as a major income generator in many small, developing island states [1]. A study by the World travel and Tourism Council (WTTC) in 2019 reported that tourism in Africa was worth approximately $29.6 billion USD (1/3 of total African tourism GPD), employing 3.6 million people. Southern Africa is one of the world’s most frequented destinations for tourists seeking encounters with wild animals [2]. Benefits of wildlife tourism may include provision of employment for local people, and increased tolerance of local wildlife [3]. However, there are potentially significant negative implications for the welfare of animals involved in these experiences [4]. One of the most highly debated and controversial of these is the elephant tourism industry [5]. Asian elephants have been used for work for several thousand years and they have a significant cultural-heritage value [6]. The rise in the Asian elephant tourism industry, particularly in Thailand, relates to the timing of the international ban on logging in 1989 [7]. The tourism industry has expanded to become a significant source of income generation across Asia, particularly in Thailand [6]. In South and Southeast Asia, it is now estimated that over 3000 elephants are used in tourism [7]. In southern Africa there is no history of using trained elephants as a workforce, and the cultural heritage that is seen in Asian countries is absent [6]. Commercial venues offering elephant rides began in the 1990s. Interactive experiences with trained elephants have increased over time, with tourists offered opportunities to ‘get closer to wildlife’ [8]. World Animal Protection [9] reported that in 2015, there were at least 39 commercial elephant venues across Southern Africa, housing approximately 215 elephants. 

Activities in which tourists can engage with elephants may include observing, walking with, feeding, bathing or riding elephants [10]. In recent years a positive shift has been reported in attitudes and behaviours of tourists towards the use of elephants in tourism [11], and tourism operators are also encouraging more responsible elephant tourism (e.g., [12]). Despite this, there are still a number of welfare concerns for elephants used by the tourism industry. Reports have highlighted elephants being kept in conditions which do not fulfill basic needs and which may involve physical restrictions, such as chaining [7]. The need for improvements in husbandry, which consider the physiological and psychological needs of elephants including social, nutritional and environmental needs, has been highlighted [5]. 

Tethering restricts an elephant’s ability to make decisions about their activities and to react according to their motivational state [13], as well as more generally limiting behavioural opportunities and choices [14,15]. Recent studies have highlighted the importance of allowing elephants control and the opportunity to make choices over aspects of their environment (e.g., when to interact with conspecifics, or access particular areas of—or resources within—their enclosure) in order to provide better welfare [16,17]. Indeed, the importance of providing an element of choice so that an animal can exert some control over their environment has been identified as essential for positive welfare in a wide range of captive wildlife species [18,19,20]. Previous studies evaluating the impacts of removing overnight tethering for circus or zoo elephants found reduced stereotypies [21], an increase in comfort, social and play behaviours [22] and generally calm ‘demeanors’ once the tethers were removed [13]. As these are all behavioural indicators of positive welfare [23], this suggests that being off tethers resulted in improved well-being in these elephants. 

Historically within zoos, the predominant stated rationale for chaining elephants during the night (when keepers or caretakers were absent) was to prevent aggression between individuals [13], reduce the likelihood of injuries, and ensure subordinate elephants were not prevented (by more dominant elephants) from accessing resources and achieving quality rest [14]. Within elephant tourism camps in Asia, chaining is believed to sometimes be the only means of controlling elephants within the relatively limited space, where reinforced enclosures are felt to be neither financially nor logistically feasible [10]. In Zimbabwean tourist camps the stated rationale for tethering elephants is to ensure they are kept secure over-night and cannot enter human-inhabited areas in the surrounding landscape (Rendle-Worthington, personal communication). 

In elephant-keeping facilities, including both western zoos, and in captive or semi-captive facilities in range countries, there are a wide range of policies governing what is acceptable practice for chaining of elephants. Generally, in western zoos the chaining of elephants for long periods of time is considered unacceptable. The European Association of Zoos and Aquaria (EAZA) guidelines state that elephants should not be chained for more than 1 in 24 h, except in justified cases (e.g., loading for travel) [24]. The Association of Zoos and Aquariums (AZA) guidelines state that elephants must not be routinely tethered for more than 2 h in 24 [25]. In Thailand, the Elephant Care Manual for Mahouts and Camp Managers provides details on acceptable lengths of chains (it states that elephants should be kept on chains that are between 20 and 30 m in length, although shorter chains are considered acceptable in confined areas [26]); however, these is no stated restriction on duration of chaining within a 24-h period. Prior to the current study, in Zimbabwe, captive elephant managers were guided by the Domesticated Elephant Management Association of Zimbabwe (DEMAZ), but there were no guidelines given in relation to chaining [27]. 

There is substantial evidence from the zoo literature that chaining elephants for long periods (and the related limited freedom to move) has detrimental impacts on welfare, including increased foot problems due to damp conditions caused by a buildup of faeces and urine in the area in which elephants are held [28], increased risk of arthritis due to the reduction in movement [29] and development of stereotypies [21,22,30]. Despite these known problems, chaining or tethering elephants is still relatively routine practice in the Asian elephant tourism industry. Across elephant tourist camps in northern Thailand, Bansiddhi et al. [10] found that 82% of camps chained elephants for a ‘significant portion of the day’, as well as over the night time period, with only 9% of camps chaining their elephants only at night and only 9% of camps not chaining their elephants at all. A recent review of elephants in six Thai camps found elephants were chained for long periods of time: a minimum of 16–18 h overnight [31]. In a review in 2015, WAP [9] highlighted that although most captive elephants in Africa have relatively unrestricted foraging opportunities in the daytime, overnight they are kept chained in small enclosures. Elephants in UK zoos were identified as being at greater risk of performance of stereotypies if they were housed in smaller overnight enclosures, especially when the difference between daytime and night-time space access was large [32]. Within southern African facilities, where elephants are tethered, this primarily happens overnight, which is approximately 12 h (from 17:30–18:30 until first light, approximately 06:00) (Rendle-Worthington, personal communication). 

Recent research by the UK charity Wild Welfare highlighted the potential for what they termed ‘developing country’ zoos to have reduced welfare standards, which may be attributed to lack of awareness of current approaches and advances in animal welfare science, and potential cultural differences in perception of animal needs [33]. It is possible that there are similar challenges in tourist camps, where access to new developments in welfare science may not be readily accessible. In recent years, there have been efforts by researchers and animal welfare scientists to work with elephant tourism operators in southeast Asia and southern Africa to engage in sharing of knowledge and expertise [34], and to co-develop evidence-based elephant management guidelines that are designed to provide positive elephant welfare [5,35]. However, whilst the outputs of this work in the Asian elephant tourism industry are well published (e.g., [5,10,36]), there is a paucity of published literature on elephants from southern African facilities. 

EleCREW is a non-profitable charitable trust which aims to improve the lives of working elephants across southern Africa [37]. EleCREW care for a herd of 10 semi-captive African elephants which live at the Jafuta Reserve in Victoria Falls, Zimbabwe. The reserve is financially supported by the ‘elephant experiences’ that they provide to tourists. When they are not interacting with tourists, the elephants are able to move freely around the 3000 ha site. Until recently elephants were kept on 4 m tethers overnight (approximately 12 h a day). However, in 2018, a new management strategy was developed at the Jufata Reserve with the aim of improving elephant welfare. As part of this initiative, they stopped tethering their elephants overnight and developed small overnight enclosures to house the elephants (‘bomas’, approximate sizes from 110 m^2^ to 310 m^2^). Within these bomas elephants could move freely, and some (socially compatible) elephants were housed together. Whilst there is evidence of the positive impacts of removing circus and zoo elephants from overnight tethers, to the authors’ knowledge, this is the first study to investigate this change in an elephant tourism facility in southern Africa. The aim of the current study, albeit a case study conducted opportunistically on four elephants, was a first look at quantitatively assessing the impact of these management changes on behavioural indicators of welfare in these elephants. 

## 2. Materials and Methods

### 2.1. Subjects and Study Site

Subjects were four adult male African savannah elephants (*Loxodonta africana*) (Table 1) housed at the Jafuta Reserve, Zimbabwe. These subjects were chosen as they were still on over-night tethers at the onset of the study. Another six elephants at the site had previously been moved into over-night bomas as part of improvements to routine management, but prior to the installation of video recording equipment. Video recording equipment enabled remote observations of the study’s four elephants, which did not impact elephant behaviour. 

All elephants were wild born and had been donated to eleCREW after being previously kept at a number of other facilities in the region. During daylight hours (approx. 06:00–18:00) the elephants had unrestricted access to a 3000 ha reserve which comprised a natural teak forest with waterholes, natural shade, grassland and wild fauna (including wild elephants). At first light they experienced an approximately 30-min handling session, and were administered any necessary medical treatment. Elephants were then involved in activities (including a short talk and interaction with the elephants) for community, school and tourist groups for up to 1 h per day. 

Overnight (18:00–06:00) elephants were tethered (prior to the management change) and then housed in bomas (approximate sizes from 110 m^2^ to 310 m^2^) (post management change). When elephants were tethered, they had no physical access to other elephants, however they were able to reach their trunks towards one another. When elephants were housed in bomas, E3 and E4 were housed together and other individuals had the opportunity for physical contact at their enclosure boundary, with elephants in adjacent pens. 

### 2.2. Behavioural Observations

Data were collected from observations of video recordings from December 2019 to March 2020 (Table 1) during the hours that elephants were housed in the bomas (approximately 18:00–06:00). Video recordings were made of the bomas using four motion-activated cameras (HIKVISION colour camera DS-2CE5-6D0T-IRP, IR distance: 20 m). In order to capture the impacts of management changes on elephant behaviour, data were collected at three time periods: Phase 1—tethered (December 2019); Phase 2—approximately four weeks post release from tethers (January to February 2020); Phase 3—approximately eight weeks post release from tethers (February to March 2020). 

Behavioural data were collected over three nights in each time period using focal, instantaneous sampling with a 5-min interval. Behaviours were recorded according to a pre-defined ethogram (Table 2). 

### 2.3. Data Analysis

Due to circumstances beyond the control of the researchers (e.g., failure of recording equipment) periods of data collection were not always equal across elephants or over time. Data were expressed as a proportion of total observations within each observation night to account for this and enable comparisons to be made across elephants and across conditions. Due to relatively small sample sizes, descriptive statistics are provided where appropriate. Inferential statistical analyses were undertaken in R (Version 1.1.383) [39] using the packages ‘lme4′, ‘multcomp’ [40], ‘emmeans’ [41] and ‘ggpubr’ [42]. Graphs were produced using the package ‘ggplot2′ [43]. Significance values were set at 0.05 unless corrected for multiple comparisons.

General linear models (GLMs) with quasibinomial error structures were used to investigate whether there was an effect of overnight management condition on proportion of time individuals spent engaging in a range of behaviours. Separate models were created for each behaviour. Data were split into two initial periods for analysis: (i) tethered, or (ii) housed in bomas (not tethered). Further analyses were undertaken to establish whether behavioural changes persisted over a longer period of time and whether there were individual-level differences. Data were split into three time periods: (i) tethered, (ii) approximately four weeks post-release from tethers (4 weeks post); and (iii) approximately eight weeks post-release from tethers (8 weeks post). Tukey-corrected post hoc tests were applied where appropriate. Behaviours were fitted as a response variable, with individual and condition as fixed effects. Model results are reported as model estimate (β_1_) ± SE. Spearman’s rank correlation was used to assess relationships between the proportion of time spent in standing rest and lying rest, to document whether type of rest behaviour changed as a result of being removed from overnight tethers.

### 2.4. Ethics Statement

All research protocols were reviewed by the University of Nottingham Ethics Committee approval number 3,576,220,407. Permission to conduct the study was granted by the Jafuta Reserve prior to commencement of data collection. 

## 3. Results

An overview of behavioural activity is provided in Table 3. Locomotory, positive social, maintenance, environmental interactions and abnormal repetitive behaviours (ARBs) were relatively infrequent across all conditions, accounting for on average less than 10% of observations. The most frequently recorded behaviours were standing and lying rest. 

Resting behaviour differed across the conditions. Across all elephants, proportion of time engaging in lying rest was greater when elephants were not tethered than when they were tethered (1.0858 ± 0.4209, z = 2.580, *p* < 0.05). Proportion of time engaged in lying rest was greater in the first four weeks post release from tethering than when tethered (1.4141 ± 0.4859, z = 2.910, *p* < 0.05) (Figure 1). There was no difference in proportion of time engaging in lying rest between the time when elephants were tethered and two-months post release from tethering (*p* > 0.05) or between one and two-months post tethering (*p* > 0.05). There was no significant difference in proportion of time engaging in lying rest between individuals (Figure 1). 

Across all elephants, there was no difference between proportion of time spent engaging in standing rest when not tethered as compared to tethered (*p* > 0.05). Proportion of time spent engaging in standing rest was greater two months post release from tethers than one-month post-release from tethers (1.231 ± 0.4275, z = 2.627, *p* < 0.05). There were no significant differences in proportion of time spent engaging in standing rest between elephants when they were on tethers or the first month post-release from tethers. In the second month post-release from tethers, E4 engaged in significantly more standing rest than when he was tethered (−3.3377 ± 0.780, *p* < 0.01) and in the first month post-release from tethers (−2.7386 ± 0.663, *p* < 0.01) (Figure 2). During the second month post-release from tethers, E4 also engaged in greater proportions of standing rest per night than E1 (−2.5290 ± 0.633, *p* < 0.01), E2 (−2.1226 ± 0.590, *p* < 0.05) and E3 (−2.3672 ± 0.614, *p* < 0.01). Across all elephants, there was a negative correlation between proportion of time spent in standing rest versus lying rest (R_s_ = −0.53, *p* < 0.001).

Across all of the elephants there was no significant difference in proportion of time engaging in positive social interactions when tethered or not tethered (*p* > 0.05). Proportion of time spent engaging in positive social interactions was also not significantly different across the three conditions (*p* > 0.05). However, significant differences were observed between elephants (Figure 3). E3 engaged in positive social interactions for greater proportions of time than E1 (1.5027 ± 0.5864, z = 2.658, *p* < 0.05) and E2 (2.0159 ± 0.7011, z = 2.875 *p* < 0.05). E4 also engaged in positive social interactions for greater proportions of time than either E1 (1.8899 ± 0.5490, z = 3.443, *p* < 0.01) or E2 (2.4032 ± 0.6879, z = 3.493, *p* < 0.01). Elephants were not able to engage in tactile contact when tethered; however, ‘reaching towards’ another elephant was observed. After they were put into bomas, E3 and E4 were housed together. Other elephants had opportunities for tactile contact at the enclosure boundary.

Across all elephants, there was no difference between proportion of time spent engaging in locomotory behaviour when not tethered as compared to tethered (*p* > 0.05). Proportion of time spent engaging in locomotory behaviour was greater in the period two-months post release from tethers than when tethered (1.1803 ± 0.4733, z = 2.494, *p* < 0.05). Differences were also recorded between individuals: E4 engaged in a greater proportion of locomotory behaviour than E3 (1.7432 ± 0.6573, z = 2.652, *p* < 0.05) (Figure 4). 

Proportion of time engaging in feeding, maintenance and standing behaviour did not differ significantly in frequency between the tethered and non-tethered conditions (*p* > 0.05) or across the three time periods (*p* > 0.05). No differences were observed between individuals (*p* > 0.05). Negative social interactions were not modelled due to low occurrence. Negative social interactions were only observed during two scans of one individual (E3) during one evening in condition 3; two months post-release from tethers. Proportion of time spent engaging in ARBs did not significantly differ when elephants were tethered or not tethered, between the three time periods or between individuals (*p* > 0.05). However, although not statistically significant, it is noted that reductions were seen in performance of ARBs, across all elephants, when elephants were no longer tethered (Figure 5). 

## 4. Discussion

The aim of this study was to evaluate the behavioural impacts resulting from the removal of overnight tethers from four bull elephants housed in semi-captive conditions in a Zimbabwean tourist camp. Elephants showed behavioural indicators of improved welfare as a result of the management change, including increased lying rest, reduced stereotypies and increased social interactions. However, there was also individual variability observed between the bulls. These responses are comparable to research monitoring behavioural changes in zoo and circus elephants after they were removed from tethers (e.g., [15,44,45]). 

The greatest behavioural changes in the current study were seen in resting behaviour after elephants were no longer tethered. In all three time periods, elephants spent more time engaging in resting behaviours than any other behaviour, which is comparable with activity budgets reported for elephants in zoos (e.g., [46,47,48]). Proportion of lying rest was significantly greater when elephants were no longer on tethers. With the exception of elephant 4, all elephants showed greater frequency of lying rest in the two periods when they were not tethered, as compared to when they were on tethers. E4 initially doubled his frequency of lying rest (during observations within the first 4 weeks post release from tethers) however during observations in the second month post-release from tethers the elephant did not engage in any lying rest. At this time E4 increased standing rest (mean proportion of overnight activity = 0.71, as compared to 0.08 when tethered and 0.14 four weeks post release from tethers) and also increased locomotion (mean proportion of overnight activity = 0.08, as compared to 0.01 when tethered and 0.02 four weeks post release from tethers). 

Whilst no studies have defined optimal levels of rest for elephants, its importance has been increasingly recognised in recent years [46,47,48,49,50,51]. Elephants must lie down to engage in delta wave and rapid eye movement (REM) sleep, phases known for their restorative properties [52], and importance for optimal cognitive function [53]. Standing rest in elephants has been described as a ‘filler’ or ‘substitute’ when adequate lying rest cannot be obtained [46,50,54]. In this study, a negative correlation was observed between standing and lying rest, suggesting that elephants may trade off between these two types of rest. Similar correlations were observed in zoo-housed Asian elephants [48]. Zoo elephants engage in lying rest daily, with multiple bouts of lying rest per night [47,55]. However, reports of wild elephants indicate that they do not always engage in lying rest daily, and instead may only engage in recumbent rest every three to four nights [56]. Researchers have suggested that recumbent rest will only occur in captive elephants if they are comfortable and feel safe in their environment [48,51,57], with both the physical (e.g., soft, malleable surfaces or sand mounds) and social environment (e.g., access to compatible conspecifics) impacting on recumbence behaviour [46,50,51]. Floor space has also been identified as a potential factor which impacts on resting behaviour in captive elephants. In a study of zoo-housed African elephants in the U.S., Holdgate et al. [47] found that time spent engaging in lying rest was highest in individuals who had the largest amount of floor space. It is unknown whether it is the space itself that is important, or the opportunity for choice in the social environment. In this semi-captive environment, there could also be a number of other external factors (e.g., external disturbances or environmental stimuli such as noises from other wildlife, or ambient temperature and relative humidity), which have been known to impact resting behaviour in wild African elephants [56]).

In the current study, once elephants were taken off their tethers, those elephants who had been identified by handlers as socially compatible (*n* = 2) were housed together overnight in bomas (Rendle-Worthington, personal communication). Positive social behaviours more than doubled in these elephants once they were released from tethers. These behaviours included tactile contact with the trunk, and leaning on one another. One of the elephants who was lone housed also engaged with trunk touching behaviour with conspecifics who were accessible at the enclosure boundary. Aggressive behaviours were extremely rare, with only two incidents being recorded in one elephant (E3) during one observation night (period 3). This aggressive behaviour may, however, have impacted on lying rest in E4. The reduction in lying rest alongside increased locomotion in E4, and the two occurrences of aggression from E3, could signify a problem with the social environment during that period. Zoo elephants display fluidity in their positive and negative social interactions, and the importance of incorporating an understanding of social dynamics in social management decisions of zoo elephants has been highlighted [58]. The occurrence of negative social interactions is an important factor for consideration in management of these elephants, and this is potentially a concern for any captive unrelated elephants. Unrelated captive elephants certainly can and do have positive social relationships [59,60,61]; however, it is important to continually monitor elephant relationships to tailor management to the social needs of individual elephants. Incidents of aggression should be recorded, and there must be the ability to separate incompatible elephants if required for both the safety and the well-being of all elephants involved. 

Elephants have extremely complex social needs [62,63] and this makes it challenging to care for them in captive environments [64,65]. The importance of social partners for both male and female elephants has been highlighted [66], but these social partners must be compatible [16]. The opportunity for captive species to exert choice or control in relation to social environments is also critical in promoting positive affective states [67]. Providing captive elephants with overnight housing of an adequate size, which enables appropriate access to potential social partners, has been identified as important for welfare of zoo elephants [15,23], and this should also be further explored for the overnight management of elephants used in tourism. 

Although not a statistically significant finding, all elephants reduced the frequency with which they performed stereotypies when they were no longer on tethers. Reductions ranged from 40% to 70%, with one elephant not seen performing any stereotypies by the second month of observations. Abnormal repetitive behaviours, or stereotypies, are behaviours with no apparent function [68]. Stereotypies have been identified as coping mechanisms, and their expression is taken to indicate either past or current stress experienced by the animal [69]. In elephants, as in many other species, a reduction in frequency or intensity of stereotypies has been interpreted as an indicator of an improved welfare state [23]. The changes in ARBs in the elephants in this study were thus interpreted as potentially indicating an improved welfare state as a result of the management change, the removal of the tethers. 

It is important to highlight that this project was a brief case study, based on a relatively small sample of bull elephants. However, it is, to the authors’ knowledge, the first study which has evaluated evidence of behavioural changes that resulted from the removal of overnight tethers for elephants held in a southern African tourist camp. The observed elephants were typical of those used within the tourism industry in Southern Africa: unrelated wild-caught adults who had been captive for a number of years. Work has already begun to introduce evidence-based management to elephants in tourism facilities in both Africa and in Asia [5,35]. This brief case study provides further evidence indicating positive behavioural impacts from increased freedom of movement overnight by removing elephants from tethers. Importantly it provides initial evidence that it may be possible to create safe and secure over-night housing for elephants which allows greater freedom of movement and greater opportunity for social interactions, in the context of tourist facilities in southern African range countries. After this research project was conducted, a new set of guidelines, which superseded the DEMAZ guidelines, was created to establish standards for the management and welfare of elephants in human care in Southern Africa. Based on the results from the current project, taken with similar findings from prior research within zoological facilities (e.g., [13,14,30], a section was included in these guidelines which stated that ‘efforts should be made to move away routine use of tethers’ [70]. It is important to note that there are a number of factors which may impact the ability of facilities to house elephants in less restrictive settings (e.g., off tethers) [10]), however, wherever possible, facilities should be supported in undertaking these management changes, in order to improve the welfare of elephants in southern African tourist facilities. 

### Recommendations for Future RESEARCH and Advancements in Practice

It is acknowledged that whilst the behavioural changes recorded suggest positive changes to welfare following removal from tethers, data collection was relatively limited (on a small number of bull elephants at a single facility, and over a fairly short period of time) and thus the results represent only a snapshot in time for this small sample of elephants. Elephants have very strong social bonds with other elephants, and appropriate social opportunities are imperative for captive elephant welfare [71,72,73]. Within zoos, it is recognised that social management of elephants should consider providing appropriate opportunities for socialisation, but also choice over when to interact with or avoid conspecifics [24]. The importance of social interactions for bull elephants has recently been recognised [74,75,76]. However, bull elephants may differ from females in their social and environmental needs, and this needs to be taken into consideration when designing social management. We thus advocate that this research is undertaken at other facilities in southern Africa, over a longer period of time (pre and post management change), and assessing both males and female elephants to determine the potential wider applicability of this work. 

Whilst behavioural changes indicative of improved welfare were seen in association with being released from overnight tethers, the elephant enclosures were still very small in comparison to overnight zoo enclosures, and particularly in comparison to their daytime space. In the UK zoo elephant population, those elephants observed engaging in the highest levels of overnight stereotypies were those for whom the difference between their daytime (larger) and night-time (smaller) enclosure sizes was largest [32]. The area in which elephants are housed and complexity of those areas, during the day and at night, have been identified as important for captive elephant welfare [16,72]. However, there is no clear understanding of precisely how much space is ‘enough space’ for captive elephants [77,78], and more research on minimum space requirements has been advocated [38]. We recommend this as a further area of research for elephants in southern African facilities, alongside discussions with stakeholders on practicalities of managerial changes discussed in this paper, in order to support facilities in improving elephant welfare whilst also maintaining human safety. Where facilities are still tethering elephants overnight, we recommend consideration of more welfare-friendly alternatives, in line with new management guidelines [70]. 

Finally, research has highlighted the importance of understanding individual differences and social relationships in zoo-housed elephants, and incorporating these into elephant management decisions in order to optimise their welfare [20,24,38,79,80]. Individual level differences were seen in this study population. We would therefore advocate that such information is similarly considered when making overnight management decisions for captive elephants in southern African elephant tourism camps. To further understand the impact of management protocols on elephant behaviour we recommend facilities undertake routine behavioural monitoring, and where possible concurrent evaluation of physiological welfare parameters. This will support the development of evidence-based practice within the industry and will complement work which has been undertaken by researchers studying elephants in Asian elephant camps [5,10,36]. 

## 5. Conclusions

There are a large number of elephants in tourist facilities across Asia and southern Africa. They are a vital source of income for local communities, but a number of concerns have been raised about the welfare of the elephants in these camps. One area of significant concern is the use of chains or tethers overnight, which restrict elephant movement and limit the elephant’s overnight behavioural choices. While it has been asserted that these tethers are in place for the safety of elephants and of local villages (to prevent elephant escapes), there are welfare implications for this practice. Studies of elephants in zoos and circuses have highlighted positive welfare impacts following the removal of these tethers. In the current study, behaviour was observed in four semi-captive bull elephants in a Zimbabwean tourism camp when they were removed from overnight tethers and placed in (relatively small) enclosures. Behavioural changes were indicative of improved welfare (e.g., reduced stereotypies, increased lying rest and increased positive social interactions). Aggression was observed in one elephant however this was minimal. Elephants did not escape the bomas nor were there any impacts on human safety as a result of the management change. However, this study was based on a small sample size at one facility with an opportunistic study design. We advocate further investigating this issue across a wider range of southern African elephant tourism facilities, in order to determine the applicability of these results across these types of facility. More generally, we suggest that routine and standardised behavioural assessments should be undertaken to evaluate the welfare impacts of changes in management and husbandry of tourism elephants within southern Africa. This approach will help to support an evidence-based approach to improving welfare of elephants within southern African tourist facilities, which will complement work which has been undertaken by researchers studying elephants in Asian elephant camps. 

## Figures and Tables

**Figure 1 animals-12-01933-f001:**
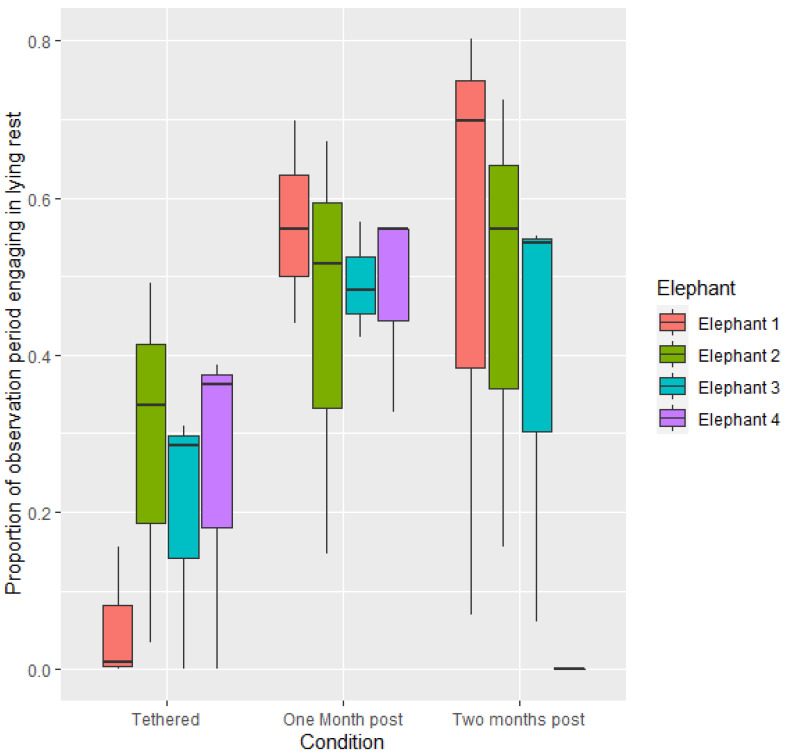
Mean proportion of time spent engaging in lying rest per observational period during the three conditions.

**Figure 2 animals-12-01933-f002:**
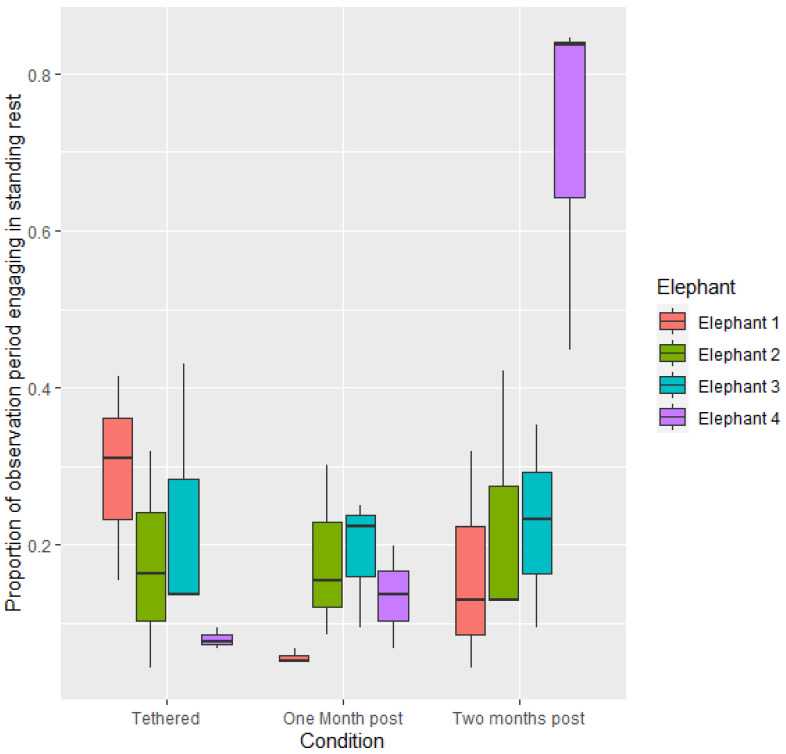
Mean proportion of time spent engaging in standing rest per observational period during the three conditions.

**Figure 3 animals-12-01933-f003:**
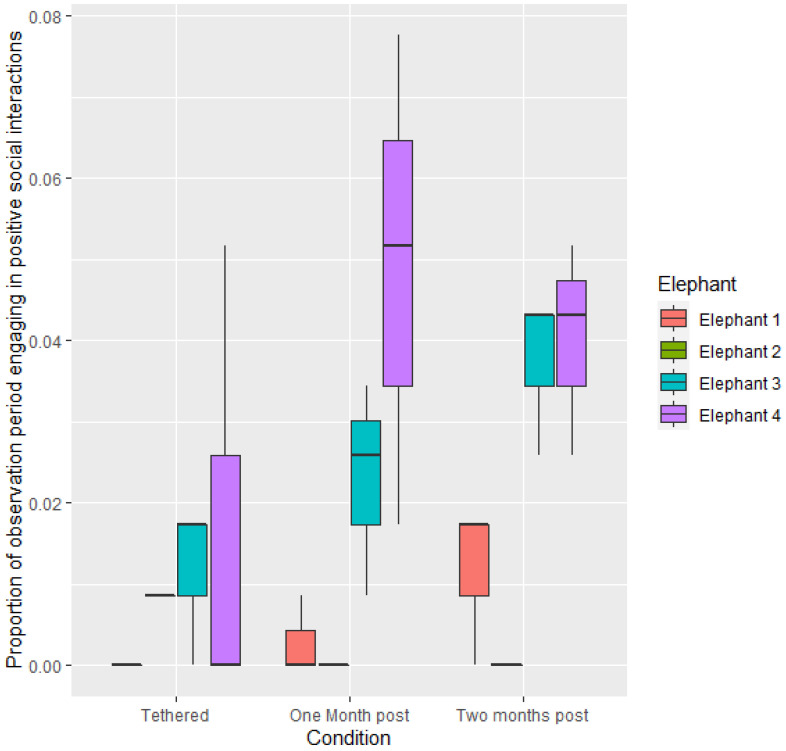
Mean proportion of time spent engaging in positive social interactions per observational period during the three conditions.

**Figure 4 animals-12-01933-f004:**
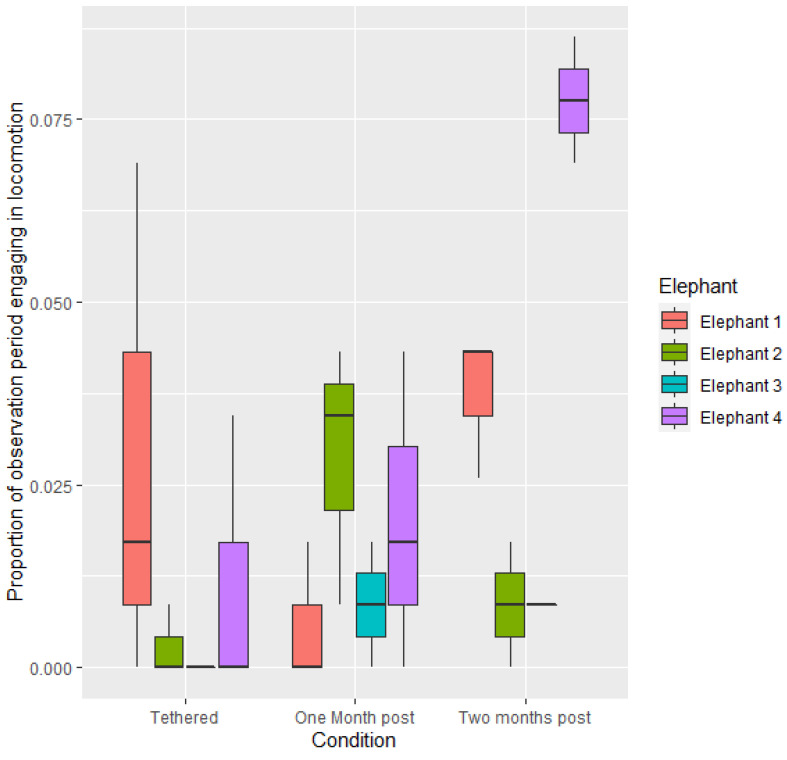
Mean proportion of time spent engaging in locomotion per observational period during the three conditions.

**Figure 5 animals-12-01933-f005:**
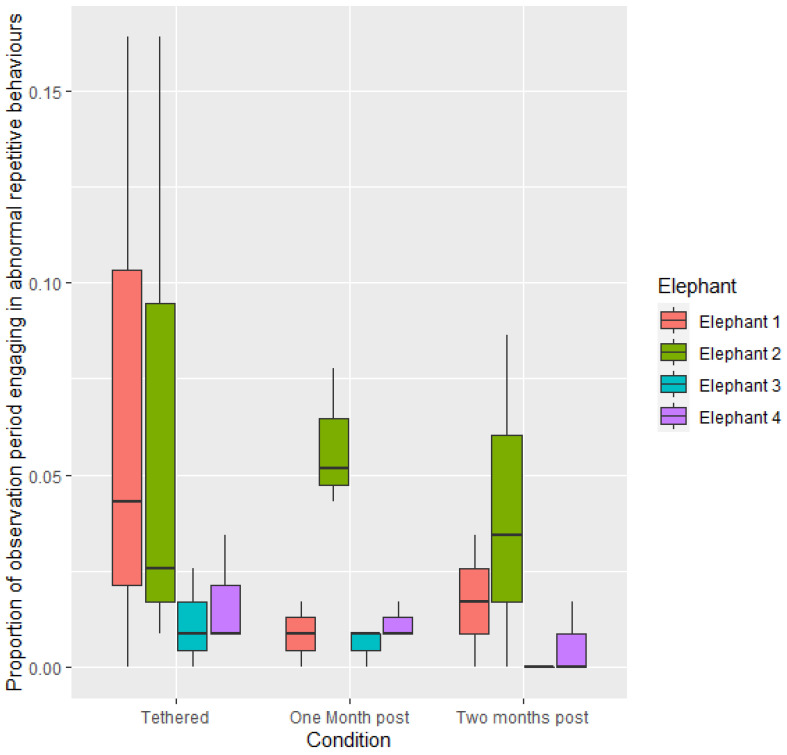
Mean proportion of time spent engaging in abnormal repetitive behaviours per observational period during the three conditions.

**Table 1 animals-12-01933-t001:** Details of study subjects and periods of data collection.

Elephant	Sex	Age at Time of Study	Housing	Study Dates			Minutes of Footage Per Condition			Mean (±SD) Number of Scans Out of Sight		
				Tethered	One-Month Post	Two Months Post	Tethered	One-Month Post	Two Months Post	Tethered	One-Month Post	Two Months Post
E1	M	19	Lone housed	24, 26, 29 December 2019	2–4 February 2020	19, 23, 28 February	1210	1355	1910	0 ± 0	4 ± 3	14 ± 1
E2	M	19	Lone housed		14–16 February 2020	1–3 March 2020	1145	1925	1965	0 ± 0	22 ± 2	20 ± 20
E3	M	45	Pair housed once released from tethers		31 Jan–2 February 2020	19, 23, 28 February	1190	1565	1745	0 ± 0	0 ± 0	0 ± 0
E4	M	23			31 January–2 February 2020	19, 23, 28 February	1190	1575	1910	20 ± 17	3 ± 2	0 ± 0

**Table 2 animals-12-01933-t002:** Ethogram of behaviours [38].

Behaviour	Description
Locomotion	Taking two or more steps in any direction in a non-repetitive pattern
Stand	Standing still but not resting, animal is alert and eyes are open
Standing rest	Upright and stationary with 3 or 4 feet on the ground. Not performing any other behaviour. Eyes may be closed. End of trunk usually curled on ground. Individual may be leaning on an object (e.g., enclosure bars, or a tree) or conspecific.
Lying rest	Lateral recumbence, no other behaviours are being performed
Social positive	Engaging in positive social behaviours (e.g., reaching the trunk towards other elephants, social play, trunk touch)
Social negative	Engaging in negative social behaviour (e.g., fighting, aggression)
Abnormal repetitive behaviour	Repetitive behaviour with no obvious function or purpose
Feeding	the process of locating and consuming food stuffs
Maintenance	Any self-maintenance or grooming behaviour (e.g., dust bath, rubbing)
Environmental interaction	Investigating or interacting with things in the environment (other than food)

**Table 3 animals-12-01933-t003:** An overview of changes in mean proportion of time per observation night spent engaging in behaviour across the three conditions, reported as mean ± SD.

Behaviour	Elephant 1			Elephant 2			Elephant 3			Elephant 4		
	Tethered	4 Weeks Post Release	8 Weeks Post Release	Tethered	4 Weeks Post Release	8 Weeks Post Release	Tethered	4 Weeks Post Release	8 Weeks Post Release	Tethered	4 Weeks Post Release	8 Weeks Post Release
Feeding	0.12 ± 0.08	0.07 ± 0.04	0.12 ± 0.02	0.05 ± 0.04	0.13 ± 0.03	0.13 ± 0.04	0.15 ± 0.1	0.13 ± 0.07	0.13 ± 0.04	0.11 ± 0.13	0.13 ± 0.06	0.2 ± 0.03
Stand	0.08 ± 0.03	0.04 ± 0.02	0.07 ± 0.03	0.07 ± 0.04	0.07 ± 0.03	0.04 ± 0.01	0.06 ± 0.03	0.05 ± 0.01	0.18 ± 0.13	0.03 ± 0.02	0.06 ± 0.04	0.05 ± 0.02
Standing rest	0.29 ± 0.11	0.06 ± 0.01	0.16 ± 0.12	0.18 ± 0.11	0.18 ± 0.09	0.23 ± 0.14	0.24 ± 0.14	0.19 ± 0.07	0.23 ± 0.11	0.08 ± 0.01	0.14 ± 0.05	0.71 ± 0.18
Lying rest	0.05 ± 0.07	0.57 ± 0.11	0.52 ± 0.32	0.29 ± 0.19	0.45 ± 0.22	0.48 ± 0.24	0.20 ± 0.14	0.49 ± 0.06	0.39 ± 0.23	0.25 ± 0.18	0.48 ± 0.11	0 ± 0
Locomotion	0.03 ± 0.03	0.01 ± 0.01	0.04 ± 0.01	0 ± 0	0.03 ± 0.01	0.01 ± 0.01	0 ± 0	0.01 ± 0.01	0.01 ± 0	0.01 ± 0.02	0.02 ± 0.02	0.08 ± 0.01
Social positive	0 ± 0	0 ± 0	0.01 ± 0.01	0.01 ± 0	0 ± 0	0 ± 0	0.01 ± 0.01	0.02 ± 0.01	0.04 ± 0.01	0.02 ± 0.02	0.05 ± 0.02	0.04 ± 0.01
Social negative	0 ± 0	0 ± 0	0 ± 0	0 ± 0	0 ± 0	0 ± 0	0 ± 0	0 ± 0	0.02 ± 0.01	0 ± 0	0 ± 0	0 ± 0
Maintenance	0.01 ± 0.02	0 ± 0	0.01 ± 0	0 ± 0	0 ± 0	0.01 ± 0	0 ± 0	0.01 ± 0.01	0.04 ± 0.03	0.01 ± 0.01	0.01 ± 0	0 ± 0
Environmental interaction	0.03 ± 004	0 ± 0	0 ± 0	0.01 ± 0.01	0.02 ± 0.01	0.01 ± 0.01	0.01 ± 0.02	0 ± 0	0 ± 0	0.01 ± 0.01	0 ± 0	0.01 ± 0
Abnormal repetitive behaviour	0.07 ± 0.07	0.01 ± 0.01	0.02 ± 0.01	0.07 ± 0.07	0.06 ± 0.01	0.04 ± 0.04	0.01 ± 0.01	0.01 ± 0	0 ± 0	0.02 ± 0.01	0.01 ± 0	0.01 ± 0.01

## Data Availability

Data available upon reasonable request from the corresponding author.

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
