# Peer review of "Behaviour and Welfare Impacts of Releasing Elephants from Overnight Tethers: A Zimbabwean Case Study"

_animals, 2022, doi:10.3390/ani12151933_

Round 1

Reviewer 1 Report

The authors have addressed an interesting topic with high relevance for animal welfare, as well as for human-animal interaction. For this purpose, they observed 4 male elephants that were transferred from an overnight tethering system to a free housing system. Two animals were kept individually and two other animals were kept together. Overall, therefore, this article appears to be more of a case report than a planned study. In my opinion, the small number of animals does not justify a statistical but a merely descriptive evaluation. Therefor I recommend a major revision. 

In my perception, the conclusions do not fully reflect the observations described. Thus, two aggressive events were recorded within the 6-day observation period, which for my feeling was not sufficiently acknowledged. The interpretation of the reduced resting while lying down for E4 could be discussed more critically. 

All in all, the article is an interesting case description, which gives valuable hints for a rethinking of the currently common husbandry practice of elephants in a touristic context. 

For any elaboration or modification of recommendations, I would, first, include a larger number of animals in a study and, second, extend the observations over a longer period of time to ensure human and animal safety. 

The text should be revised for punctuation as well as harmonization of spellings in British English. 

Reviewer 2 Report

Thank you for the opportunity to review this paper by Williams et al. entitled, "Releasing elephants from overnight tethers: impacts on behaviour and welfare." The paper focuses on a small sample of four male African elephants living in a facility in Zimbabwe, and presents data suggesting that the untethering of elephants improves their welfare. I found the paper to be well-written (but see below for some needed proof-reading), and generally well-presented. My biggest concern is that the paper is written simultaneously as a "case study" (the authors' own words, and I agree) and as a wide-ranging study with broader implications for elephant captivity everywhere. The authors should be much more conservative in their discussion of the study's implications, considering that the elephants in their sample (N=4) are housed at a very unique facility in Africa, with limited scope for extrapolating to other populations either in Africa or Asia. The authors never differentiate between the captive circumstances in Asia and Africa, which are markedly different, nor the socio-economic differences within and between the two continents in terms of tourism pressures or captive elephant management. In general, I strongly encourage the authors to expand their discussion of the differences in captivity in Africa and Asia, but contain their discussion of their study's implications to the former. Below I provide some minor comments on the manuscript, which the authors can easily address, but in general, I strongly encourage the authors to focus on the data they have; this is a case study on four elephants in a single facility, and the implications of this study are far more relevant for the care of elephants in similar facilities in Africa than they are for captive elephants in Asia.

1 - The paper has typos and several grammatical errors - without listing them here, I encourage the authors to carefully proof-read the paper and revise prior to next submission.

2 - In the introduction, all of the background on the care and maintenance of captive elephants in Zimbabwe is cited as the personal communication from one of the co-authors. Is there no literature that can be cited here? 

My biggest concern with this paper, and it starts in the introduction, is that, while the authors briefly mention the limited sample size and the fact that this paper should be considered as a case study, they also make broad stroke claims about the implications of this work to the care of captive elephants everywhere (case studies should never be presented as representative of an entire population, so the authors really need to tone down their language, from the title, which implies this is a far-reaching study with a large sample, through to the discussion).

As I stated before, the authors should differentiate more clearly between research on African and Asian elephant captivity. The authors never mention the fact that, in Africa, space for untethering elephants is, in general, far more plentiful than it is in most Asian range countries. While mahout hesitancy is a major factor preventing the improvement of elephant welfare practice in countries such as Thailand, India and Sri Lanka, it is also true that most elephant owners in these countries have very limited space and financial resources in order to provide less tethering and greater space opportunities for their elephants. To address this, I suggest the following:

The authors should emphasize in the intro, as they do in the discussion, that this research is indeed a case study. Thus, the authors should clarify that this study may be applicable only to elephants living in conditions similar to those in this very specific sample. The results from four MALE elephants untethered for limited periods of time at a single location in Zimbabwe can not be extrapolated to entire populations of captive elephants living in other countries. The authors should also acknowledge the numerous factors impacting the welfare practices of elephants in other countries, and make it clear that the circumstances under which captive elephants in Africa live are considerably different than those of elephants living in Asia.

METHODS

1 - Why were only (4) elephants tested if the facility has 10? Why only males? More details on the sample and the availability of other elephants in the sample is required. Also, are these elephants wild- or captive-born? This is also relevant to the larger discussion of the study's applicability to captive elephants at large. Male behavior is markedly different than female behavior, especially in terms of their sociality, and when you discuss the 'extremely complex social needs' of elephants, the sexual differences should also be discussed. While male elephants do indeed have social needs as well (both with other males and with female-led family groups), I think a discussion of the variation in sociality between sexes is relevant here. 

2 - The elephants were kept in 110m2-310m2 bombas), but what were their normal daily activities like? Did they have access to larger areas during the day? Did I miss details about how often they were tethered normally and for how long? Did this vary within the population? Elephants in Asia may be tethered at night but many walk long distances with their mahouts during the day, so again, more details are needed here.

DISCUSSION

1 - “And [the expression of stereotypies] is taken to indicate either past or current stress experienced by the animal.” Many papers make this argument but I have never seen an empirical investigation of this - if there is one, cite here. If there isn’t one, this language is unwarranted.

2 - “The observed elephants were typical of those used within the tourism industry..” Contain this to Africa; the elephants in your sample are not representative of those in Thailand, as most were not wild captured. Unless I missed it, did you mention whether the elephants in your sample were wild caught or captive born? That is an important point, add to Table 1.

With a reframing of the intro and discussion to emphasize the limited focus of this study and its implications, I think it warrants publication in Animals.

Reviewer 3 Report

The manuscript is well written and clear to understand. It is an interesting and worthwhile study. The sample size is small so may not allow for between individual comparisons. It would be good if the authors can provide history of the elephants (e.g. were they rescued species?).  They can also provide more detail in terms of how they can advocate the findings to the elephant tourism sector locally to make sure that the welfare of the elephants is looked after.

Round 2

Reviewer 2 Report

Thank you for addressing my concerns.